# *'We do not talk about it'*: Engaging youth in Malawi to inform adaptation of a mental health literacy intervention

**Sandra Jumbe**[1,2]*, **Joel Nyali**[2], **Maryrose Simbeye**[3], **Nelson Zakeyu**[4], **Gase Motshewa**[2], **Subba Rao Pulapa**[2]

**1** Centre for Evaluation and Methods, Wolfson Institute of Population Health, Queen Mary University of London, London, United Kingdom, **2** Millennium University, Blantyre, Malawi, **3** The National Youth Council of Malawi, Lilongwe, Malawi, **4** Drug Fight Malawi, Lilongwe, Malawi

* s.jumbe@qmul.ac.uk

## Abstract

### Purpose

There is limited knowledge on how to tackle mental health problems among youth in Africa. Literature describing community engagement (CE) approaches in low/middle-income countries (LMICs) health research is sparse. CE with youth from LMICS can help steer and shape culturally relevant interventions for stigmatised topics like mental health, resulting in better healthcare experiences. We share our experience of engaging youth in Malawi through advocacy organisations to inform cultural adaptation of a mental health literacy intervention.

### Methods

Young people were recruited using social media from universities and community youth organisations in Malawi to participate in focus group discussions to help culturally adapt content of an existing mental health literacy intervention. Nine online focus groups with 44 individuals were conducted. Discussions involved views and experiences of mental health, including impact of the coronavirus pandemic. Discussions were recorded, transcribed verbatim and analysed using content analysis.

### Results

Transcript analyses revealed a vicious cycle of poverty and mental health problems for youth in Malawi. Four key themes were identified, 1) poverty-related socioeconomic and health challenges, 2) no one talks about mental health, 3) lacking mental health support and 4) relationship issues. These themes fed into one another within this vicious cycle which perpetually and negatively impacted their lives. The coronavirus pandemic worsened socioeconomic issues, health challenges, mental health and substance use issues, and burden on Malawi's already weak mental health system.

**Data Availability Statement:** All relevant data are within the paper and its Supporting information files.

**Funding:** SJ was awarded funding from the Global Challenges Research Fund Large Grants Scheme from Queen Mary University of London to conduct this research. The funders had no role in study design, data collection and analysis, decision to publish, or preparation of the manuscript.

## Conclusion

Findings suggest increasing untreated mental health burden among Malawi's youth. It highlights great need to address mental health literacy using existing community structures like educational settings to minimise burden on a weak health system. Online focus groups are an effective way of acquiring views from various young people in Malawi on mental health. This CE approach has grown our stakeholder network, strengthening potential for future CE activities and broader research dissemination.

## Introduction

There is a dearth of knowledge on how to tackle mental health problems in sub-Saharan Africa, especially for young people [1]. This has coincided with an epidemic rise in substance use, not only from commonly known substances like alcohol and tobacco, but other sources of addictive legal highs such as codeine cough syrup. A recent systematic review found an overall prevalence of 41.6% for 'any substance use' among adolescents in sub-Saharan Africa [2]. This lack of information on mental health has resulted in some countries taking a criminalising approach, further disenfranchising an already vulnerable group.

Malawi, a low-income country located in Southern Africa has multifaceted development challenges [3]. Most Malawians depend on very small-scale businesses and subsistence agriculture. Moderate health facilities in Malawi are based in the cities, whereas 86% of the population live in the rural areas where health facilities are quite meagre. This is clear evidence of health inequalities with regards to access to health services. Furthermore, 80% of development expenditure for Malawi's health sector is donor supported. Most youth in Malawi (82%) live in rural areas and are exposed to disproportionate challenges such as poor-quality jobs, early marriages, and difficulties accessing healthcare [3]. Over reliance on donor support has led to focus on health issues which are poorly aligned with national health priorities, resulting in neglect of basic primary health care systems and preventive health services particularly in rural Malawi [4]. These challenges progressively impact a large proportion of the population given that more than 46% are below age 15-year-olds, and youth aged 15–29-year-olds account for a quarter of the population [3]. The social inequalities highlighted above, e.g. youth unemployment, are well known risk factors and consequences of mental health problems and substance use [5, 6]. Mental health and substance use issues are largely left untreated due to limited (mental) health services [7, 8].

Depression in Malawi is common with prevalence rates of up to 20% among adolescents [9] and 30% more broadly among young people [10]. Substance use problems can add to mental illnesses by making symptoms worse or harder to treat [11]. Alternatively, alcohol and drugs in risky ways to well-known coping mechanisms for mental illness, its symptoms and medication side effects [11, 12]. Existing studies in Malawi reveal increased problematic substance use among young people [13, 14] and early onset of drinking among adolescents [14]. However, understanding of mental health and mental illness in Malawi is very poor. Many people attribute causes of mental health disorders to alcohol and drug abuse or spirit possession, resulting in stigma, maltreatment, and discrimination towards people with mental health issues [15, 16]. The country's chronic lack of mental health services and healthcare professionals amplifies limited treatment access, knowledge, and negative attitudes [1, 7]. To address gaps identified in literature, we have conducted a pilot study that involves culturally adapting an existing

mental health literacy [17] into an e-curriculum for young people in university settings in southern Malawi. The three anticipated outcomes are enhanced understanding of mental health, decreased stigma and enhanced help seeking ability among workshop attendees [1].

This paper focuses on the intervention's adaptation phase, which involved engaging with groups of young people in Malawi (key stakeholders) in consultations to understand current contextual issues that affect their mental health [18]. Community engagement (CE) is increasingly recognised as an essential part of health research practice particularly in low-income settings or with disadvantaged groups [19, 20]. The mutual benefits of CE include learning from public groups and ensuring that one's research is relevant to society whilst raising awareness about research [20]. We report findings from focus groups with these youth groups in Malawi conducted as part of CE.

## Materials and methods

Considering the knowledge gap identified in literature regarding mental health in Malawi, we used an exploratory descriptive qualitative design [21, 22] to explore views of young people in Malawi on mental health. We hoped their insights and experiences would provide us with important information on how to adapt content of an existing mental health literacy intervention [17] for online delivery to students in university settings. Online delivery of the intervention facilitated continuity of participant learning during (physical) school closures due to the coronavirus (COVID-19) pandemic.

### Study setting

The study was conducted at Millennium University (MU) in Blantyre, Malawi. There were no restrictions to physical geographical scope because study participation was virtual due to the COVID-19 pandemic. To engage young people (aged 18–25 years), we targeted all universities in Malawi which are conveniently spread across the country, as these sites serve large youth populations from both urban and rural settings. The bulk of Malawi's (public and private) universities are based in the Southern region where 7,750,629 people live according to the 2018 national census data from the Malawi National Statistical Office [23]. The central region where the capital city is based has a similar population i.e., 7,523,340. There are two public universities in the least populated Northern region where approximately 2,289,780 people live [23]. We also targeted young people outside the university community linked to our study collaborators, The National Youth Council of Malawi (NYCoM) and Drug Fight Malawi (DFM). To ensure credibility of this study, we purposefully targeted recruitment from multiple stakeholder sources i.e., university students, recent graduates, community/youth group leaders and youth advocates, to obtain wider views of youth mental health in Malawi. Qualitative data obtained from a sample of these settings with a countrywide scope can be transferable to other youth in Malawi.

### Participant recruitment

Participants were recruited online using MU and NYCoM's Facebook pages, alongside the principal investigator's Twitter page. Specifically, a study poster advertising the research was posted on these social media platforms on 28th January 2021 (Fig 1). All universities in Malawi were tagged onto the Twitter post using their Twitter handles. NYCoM and DFM also promoted the research to their pool of existing youth (community) groups they readily engage with during organisational activities.

57 young people contacted the principal investigator (SJ) in response to the study advertisement on social media. An additional 28 potential participants came through NYCOM & DFM.

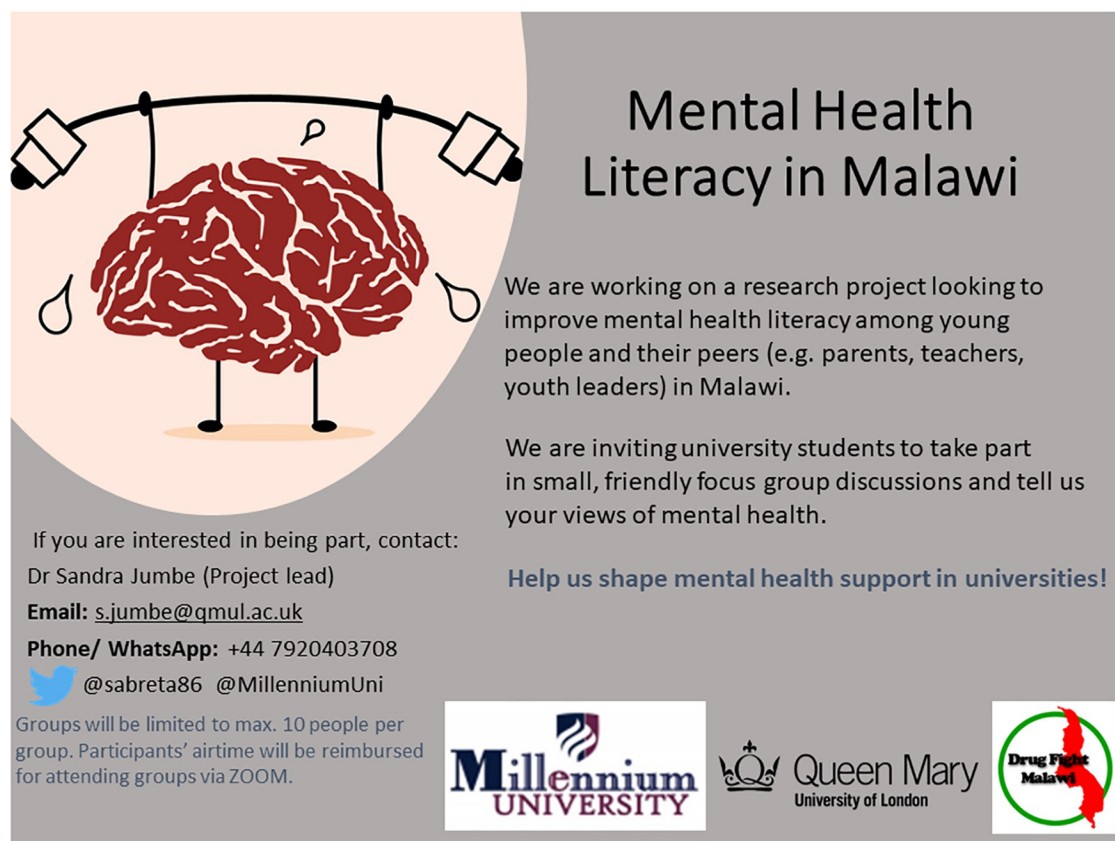

**Fig 1. Study poster advertising focus groups.**

SJ provided detailed information about the purpose of the focus groups, and mode of participation i.e., online using Zoom platform, to all responders. From these 85 responders, 75 agreed to participate and they were given a focus group date and time by SJ. 12 focus groups with between 2 to 7 participants were arranged over a 3-week period. A total of nine focus groups were conducted with 44 young people from university and youth groups using Zoom platform (Table 1). Six participants were unable to join their focus group due to network problems or technical issues with their phones/ laptops. 25 other participants who did not attend were no longer responsive when followed up.

## Data collection, analysis, and management

Focus groups lasted approximately two hours each and were all conducted by the Principal Investigator (SJ). SJ is a health psychologist and health researcher with expertise in mental health, qualitative research methodology and health intervention development. In addition, being Malawian enabled the principal investigator to use her cultural background and contextual knowledge of growing up in Malawi to draw out in-depth information during the focus group discussions on current challenges that young people face linked to mental health, needs, and available support services. She facilitated focus group discussions in both English and Chichewa (local languages in Malawi) using a guide with open-ended questions developed by the study team (Table 2). The questions were based on the study objectives, existing literature, NYCOM and DFM experience on socioeconomic issues affecting youth in Malawi. The

**Table 1. Sample description.**

| Demographics | Number (n = 44) |
|---|---|
| Male | 16 |
| Female | 28 |
| *Recruitment route* | |
| Twitter or Facebook advert | 28 |
| Local youth organisation | 16 |
| *Occupation* | |
| Student | 21 |
| Other i.e., graduates, youth leaders/advocates | 23 |
| *Malawi Region* | |
| Central | 8 |
| Northern | 6 |
| Southern | 30 |

questions were reviewed after the first and third focus groups by the study team and remained unmodified for subsequent focus groups as we were confident that they were eliciting required data. All study participants received an equivalent of £4 reimbursement to cover telecommunication and data costs incurred whilst participating in the focus groups.

Qualitative content analysis was conducted to describe young people's experiences and views of mental health care in Malawi to understand current contextual issues that affect their mental health, identify treatment gaps and stakeholders' proposed solutions for improving mental health facilitated by NVivo software. NVivo is a qualitative data analysis computer software package that helps qualitative researchers to organise, analyse and find insights in qualitative data [24]. Data analysis occurred in tandem with data collection, starting after the third focus group and in accordance with the Consolidated Criteria for reporting Qualitative research (COREQ) [25]. All audiotaped focus groups interviews were transcribed verbatim, with sections spoken in Chichewa (local language) translated into English by the principal investigator (SJ) and research assistant (JN). Both SJ and JN, who are fluent in Chichewa and English languages, listened to all audio recordings and verified the transcripts and translations from Chichewa to English.

**Table 2. Focus group questions.**

1. What are the most important issues affecting young people in Malawi at this time?

2. How do these challenges affect the mental health of young people and those in their community?

3. Alongside mental health, recent data is showing an increase in harmful substance use like binge drinking, tobacco or chamba smoking in Malawi. There have also been reports of deaths due to overdosing on hard drugs. In your view, do you think alcohol consumption or smoking is harmful?

4. Are there particular people or groups in Malawi that you think commonly use substances in a harmful way or are at risk of substance use? If yes, why?

5. How has coronavirus affected the concerns you mentioned earlier?

6. Where do Malawians most likely go when they need mental health information?

7. What kind of mental health services are needed in the community?

8. How do you guys typically obtain mental health information?

9. What would be good places to provide mental health awareness programs or support services for young people in Malawi?

**Table 3. Example of content analysis process.**

| Focus group | Meaning unit | Condensed description | Codes | Subtheme | Theme |
|---|---|---|---|---|---|
| 11 | I remember one time, my friend, we had, we are in the same class. His fees accumulated until one point several million to pay and when chatting with him, he just became like someone very low. Why? He is thinking how his school will progress. So, money issues, they cause depression | His fees accumulated. Several millions to pay. He became very low. Money issues cause depression | Financial problems Depression | High youth unemployment and lacking opportunities Depression and anxieties | Poverty related social economic and health challenges We do not talk about it |
| 6 | *unemployment is a huge huge issue in Malawi because you think you're in university or oh I'm going to get a job and then I can start building up my life, you know. Start thinking about other things you know but because of unemployment a lot of the youth are just left kind off stranded* | Unemployment is a huge issue in Malawi. Because of unemployment a lot of youth are just left stranded | unemployment | High youth unemployment and lacking opportunities | Poverty related socioeconomic and health challenges |
| 9 | looking at poverty and the way it drives people to do different things, even young people, we aren't accessing the services we ought to have coz we don't have the trading power. We don't have the finance coz of all poverty, and looking at how people act these days, it is more like… there is this element of survival of the fittest whereby you have to hustle for the day in order to get something | Poverty drives people to do different things. Young people aren't accessing services we ought to. We have no trading power. We don't have the finance coz of poverty. It is survival of the fittest. You have to hustle to get something | Poverty | High youth unemployment and lacking opportunities | Poverty related socioeconomic and health challenges |

An inductive methodological approach was used to analyse data from the focus groups based on the content of young people's thoughts and experiences regarding mental health [26]. The qualitative content analysis approach enabled us to interpret meaning from the written content of the focus group transcripts illustrated with quotes drawn from the topical area. The analysis was performed in five steps (Table 3): 1. focus group transcripts were read and re-read alongside note making by SJ and JN for increased understanding and familiarity with the content; 2. meaning units (sentences or paragraphs) linked to views or experiences of mental health were selected using an inductive approach concerning (a) young people (b) mental health care needs and solutions; 3. each meaning unit was condensed into a description of views or experiences mental health or related issues and labelled with a code; 4. subthemes were subsequently identified and grouped in relation to codes; 5. finally, an overarching finding was identified: 'vicious cycle of poverty and mental health problems' which pulled together four main themes. These (sub)themes are illustrated with quotes. Data saturation was reached after analysis of the seventh transcript.

**Rigor.** Several steps were taken to ensure trustworthiness of this qualitative work. Confirmability was ensured by including various researchers during the process of analysis [27] To ensure reliability and consistency of coding, SJ and JN independently read two transcripts line by line to identify and assign codes to similar concepts repeatedly identified from the data, then met to generate a code list based on consensus between coders. SJ then coded the rest of the transcripts. Links between these codes were identified, repeatedly identified codes were merged into (sub)categories, then themes and subthemes were generated. Any identified discrepancies were discussed with the study team/co-authors and resolved by joint consensus. Involvement of all co-authors at different stages enhanced credibility [27]. For example, the focus group questions were formulated by SJ, MS and NZ, whilst the coding scheme developed, analysis and interpretation were conducted by SJ and JN. SRP and GM conducted a critical intellectual revision of the manuscript for relevance of content. The research team comprised of expertise in psychology, community development, social science,

human resource and (higher) education. As detailed in the 'study setting', including young people from both universities and various local community settings increased transferability of our findings to other youth in Malawi [27].

## Ethics approval

This study received ethical approval from The Queen Mary Ethics of Research Committee (QMERC) on 14<sup>th</sup> December 2020 following review of the pilot study protocol and related study materials (reference QMERC2424a). The focus groups were part of CE work so research ethics and obtaining written participant informed consent was not necessary for this work. We obtained verbal consent for focus group participation and recording of the interactions from individuals prior to and at the start of each focus group.

# Results

## Main findings

The content analysis of the nine focus group discussion transcripts revealed a vicious cycle of poverty and mental health problems for young people in Malawi (Fig 2). Four key themes were identified, namely 1) poverty related socioeconomic and health challenges, 2) no one talks about mental health, 3) lacking mental health support and 4) relationship issues. These themes were key components of the vicious cycle that fed into one another and/or also perpetuated each other in a way that negatively impacted young people's lives.

## Theme 1: Poverty related socioeconomic and health challenges

We found similarities across the nine focus groups regarding key issues that are affecting the lives of young people in Malawi. During introductions, participants talked about what they did

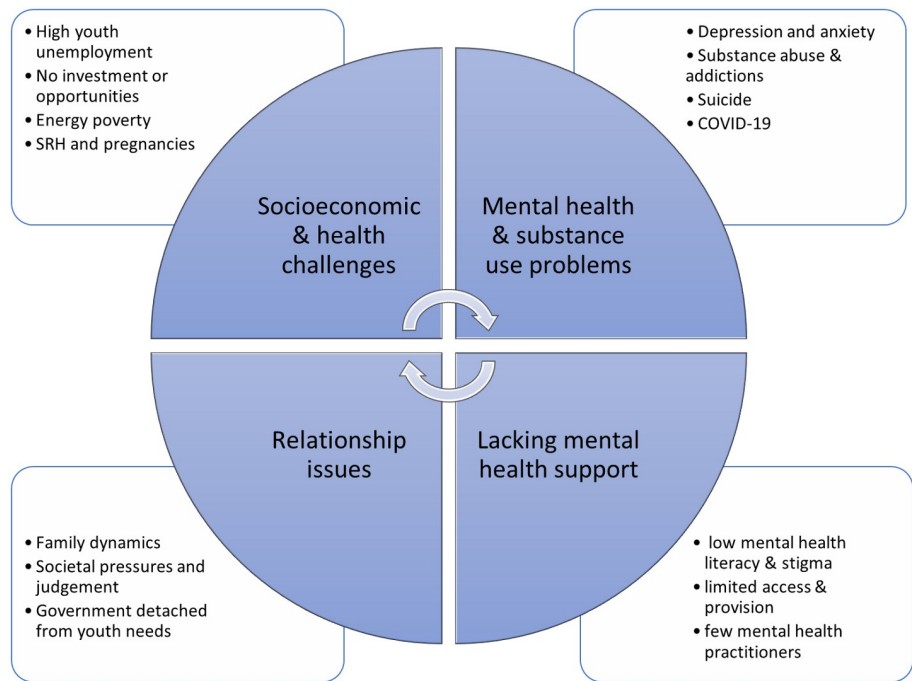

**Fig 2. Thematic map of main findings.** Bullet point around the cycle represent subthemes linked to each main themes.

in their spare time. These ranged from home-based activities e.g., reading, music, art and watching TV, to more outdoor activities like sports and festivals.

'*I like drawing, painting, reading and watching series*'

[*FG6*]

'*How I usually spend my time apart from working is hanging out with my friends and hanging out with my family. So, I guess going to concerts, shows. . . like summer jam*'

[*FG1*]

Most activities were aligned with what most young people do. However, in general, many participants highlighted a lack of youth centred activities and spaces in Malawi.

'*I will talk from my own experience sake. We don't have much spaces for young people in the country. Most of the activities young people indulge in are self-created.*'

[*FG9*]

This is a particular issue for young people in rural or less affluent areas. Participants believed this lack of activities and boredom leads youths to irresponsible behaviours, like drug abuse and risky sexual behaviours.

'*The only thing you can find in the village is one football or netball. Apart from that you find that they go indulge themselves in some other traditional dances which is also one way or the other encouraging them to drink in order to perform better*'

[*FG8*]

'*Some go out maybe because they are stressed at home they think if they go to particular places, they can meet friends and chat. At the same places they share ideas, and one of the ideas is how they are going to make themselves strong and be active in a society. Others they cheat their friends that if you drink you can be a good guy because you can approach things with a strong mind. Others, they smoke and not only smoking and drinking but others also engage in sexual activities.*'

[*FG9*]

## High youth unemployment and lacking opportunities

Unemployment and general financial insecurity were other major issues identified as affecting the youth in all the focus groups. Both current students and graduates voiced concerns around getting employment after graduating or loans to start business. Participants had ambitions to finish university, get a job and start building a life for themselves but because '*unemployment is a huge huge issue in Malawi . . . a lot of the youth are just left stranded*' [FG6]. One participant gave an example of how employers take advantage of youth by taking them on as unpaid interns on a long-term basis due to lacking work experience opportunities and high youth unemployment.

'*Another thing is that this whole Malawi internship thing, it got to a point where in Malawi they have an intern in an office. They will keep that person literally for three years (on*

*internship). Not even noticing them and their capabilities. They will not employ that person. Imagine that person can remain an intern for three years*!'

[*FG2*]

A few participants (n = 3) also raised the issue of energy poverty which further hampers the ability of youth in Malawi to develop and create wealth. Many communities do not have electricity and gas, let alone internet. Those who do spend most of their income paying extortionate bills for unreliable energy supply. Energy poverty limited young people's ability to study at home, build and/or grow affordable business.

'*We are talking of eleven out of hundred people getting electricity in the country. How could we develop in such particular nation, as a country? That's one of the problems young people are facing in this country because the barbershop, the welding shops could grow smoothly if we had reliable power in the country*

[*FG9*]

These issues that amplify financial hardship and poverty lead to mental health issues among young people.

'*You will find that because you are lacking money, your thoughts become very poor. So, what we all think is I should just drink or maybe, I should just smoke. I think my stress will go, my stress about money issues. So that's what normally happens to a lot of young people these days because they lack money*'

[*FG9*]

**Youth specific health challenges.** Participants from three of the nine focus groups voiced concerns about young people living with HIV and AIDS. Our participants stated that this group of young people faced stigma and discrimination which affects their ability to manage their medical condition. One individual mentioned that the transitional stage from adolescence to young adulthood when one goes to university is particularly challenging for this group as most are suddenly left alone and unsupported, with no close medical tracking or mentorship which brings on mental health struggles. Due to difficulties living in a society where they face constant stigma, some resort to drug and substance abuse as a coping strategy.

'*When somebody can just say 'I'm HIV positive', you see a lot of people will be going out, spreading those things. Spreading the news to other people, 'this person is HIV positive' and then people end up being discriminated and later on it depresses that person*'

[*FG6*]

In all nine focus groups, participants talked about health issues that particularly affect the youth of Malawi. One key health issue participants concurred was the COVID-19 pandemic has negatively impacted the lives of young people in Malawi, particularly with regards to mental health. COVID-19 has brought fear among young people because they are unable to live their lives like they used to. Participants reported that many people who contract COVID-19 in Malawi do not go to hospitals for treatment due to misinformation on home-based remedies and conspiracies that hospital workers are deliberately killing patients to receive more

donor funds. Study participants also described young people's fears of contracting the disease and dying, but also losing their loved ones.

'*It (COVID-19) is really affecting the mental health of many people and also we should bear in mind that many young people are losing their parents, they are also losing their friends, they are also losing the people they look up or maybe they were supporting them*'

[*FG8*]

The pandemic has also led to an increase in teenage pregnancies, sexually transmitted diseases, and early marriages, which has particularly affected girls' prospects.

'*There is an increase in teenage pregnancies and school dropouts because recently the school was opened after a period of closure. We found that the numbers were very high that did not go back to school after the first phase of the Covid 19 period in Malawi*'

[*FG8*]

Participants also felt COVID-19 had made existing socioeconomic challenges worse. For instance, lockdowns and school closures imposed by government as part of COVID-19 prevention measures led to increased school dropouts, more unemployment, underemployment and closing of businesses.

'*It's not just about people not recruiting but even big companies right now are retrenching because of COVID. Money isn't coming in as it was, and they are actually letting go of some staff because they can't afford to pay everybody*'

[*FG2*]

### Theme 2: 'We do not talk about it'

Alongside the socioeconomic and health challenges that participants described, a common phrase expressed by many participants in all nine focus groups when discussing life challenges, mental health information sources and support networks, was that people '*did not talk about it*' [*FG1*]. This unwritten societal code of non-disclosure makes young people unable to open up about personal problems which results in a hidden web of mental health and substance abuse issues, suicidal thoughts and cases. Circumstances and impacts of this phenomenon are detailed below.

**Depression and anxieties.** All participants stated that mental health is a major problem for young people in Malawi but largely unaddressed. They felt depression and social anxieties among youth were on the rise. Suggested reasons for mental distress included financial challenges, academic pressures, stigma, lacking knowledge and societal expectations.

'*Depression is something that has affected a lot of youths in Malawi nowadays based on the suicide rates that are increasing at an alarming rate. A lot of people are depressed.*'

[*FG6*]

In one focus group, several participants shared their own personal experiences of living with mental health conditions and the challenges they faced when trying to get support.

'*I can relate to experience in terms of that I don't go out and socialise either. And I think that, like, for me it's also social anxiety. I just don't like going outside, being seen, or like in public places.*'

[*FG1*]

Participants also felt one cannot talk about youth mental health without talking about social media. Issues around presenting yourself in ways that fit with Malawi's cultural expectations and cyberbullying were common topics of discussion. Although participants saw social media as a great way for youth to connect with others and get reference points on what is on trend nationally and globally, they felt these platforms also bring about pressures to fit in that can negatively impact an individual. Many participants felt that it is much easier to judge or bully someone on social media platforms because you are behind the screen.

'*We use social media to connect but I think it's turned into something toxic. And I feel like because especially the younger you are you try to fit in and maybe be part of the cool crowd or whatever the case may be. . . a lot of people are finding themselves being either a part of that or falling victim to that. So, I think right now, I'll speak personally but also generally, social media is definitely playing a role and I think it's one of those key things that's affecting mental health I think for young people.*'

[*FG1*]

Female participants especially felt that they have to be careful about presenting themselves in a way that fits Malawian conservative culture to avoid negative comments or tarnishing their reputation.

'*It's shocking and your life can be destroyed but you're only 14 years old and everybody will judge you. Your life is over. You will be labelled as this, promiscuous person. Oh, she's easy. This and that. The information is very easy to spread, so I think judgemental attitudes are getting worse*'

[*FG6*]

'*You can never post you wearing a short skirt or maybe a bikini where you went to the beach. It's like people, when you post that, they are going to come at you so hard. Like, it's your body but it is such that the culture is made in such a way that you can't dress the way you want.*

[*FG1*]'

Even though mental health problems are evident, young people feel that no one in Malawi really talks about mental health and very few people have a good understanding of what mental health is. Participants also expressed a lack of actual words in the native language to describe certain feelings, emotions, and mental health conditions.

'*In the past, I couldn't find anything serious on mental health coz one thing in Malawi that we lack is mental health literacy. Even if you go to secondary school, you can ask about mental health, it will be hard, people won't answer. Many people don't know.*'

[*FG11*]

This low mental health literacy led to persisting stigma which limited help seeking among young people, causing a lot of people to suffer in silence.

'*I feel like our generation is suffering because even if we talk to our own parents about mental health they don't understand. So, it's very hard for someone who does not even understand mental health to even pay for something like therapy for example.*'

[*FG1*]

**Hidden substance abuse & addictions.** There were many accounts from participants about young people abusing substances, with alcohol, chamba (local term for marijuana) and tobacco smoking being most common. For example, those in focus group 2 discussed the banning of very potent alcohol sachets a few years ago because of how rampant and accessible they were to young people, including children as young as nine years old. People also recounted use of less common concoctions, for instance, students from primary schools to universities mixing codeine cough syrup (Benylin) with Sprite to get high. In another group, participants talked about a new drug called 'cuba' which is becoming popular among young people in the capital city.

'*I've stayed in town. I've seen people take drugs. We talk of these, I've just forgotten the name, they are mostly sold by Nigerians here in Malawi, yeah . . .*

*SJ*: *Cocaine*?

*P2*: *Cuba*?

*P1*: *yeah, cuba, those things. That stuff, and yeah, it's taken a lot among the young people maybe from secondary school, nowadays even from primary school to the university*

Many participants felt substance use and addictions among the youth is another large problem, but because society does not address it, the issue remains a hidden epidemic that is getting worse. For many, substance abuse is a consequence of mental health issues from the many social, economic and health challenges young people are facing in the country.

'*Tobacco, cigarette smoking, the use of chamba and even alcohol is actually a major problem and a contributing factor to a lot of problems that people are facing. Because usually people go to these things thinking that maybe they are going to find some sort of relief. Maybe if they are going through another mental problem they think maybe if they can use this drug, or take alcohol, everything is going to be sorted out*'

[*FG3*]

Participants felt that a key reason young people abused substances was poor understanding and knowledge about the dangers and long-term impacts.

'*In Malawi. . . we don't' really understand about a person being an alcoholic. So, we literally have no one to tell this person when they are an alcoholic. We don't even have AA groups as we see in movies. So, if the person becomes an alcoholic, the person does not know*'

[*FG3*]

They also felt that alcohol is not viewed as negatively as tobacco or chamba smoking because it is somehow more socially acceptable, from both cultural and religious viewpoints. For example, some tribes culturally encourage early initiation of drinking among boys because it is seen as a rite of passage to manhood so heavy drinking in this context is a social norm.

'*if I take too much alcohol that is a sign that I'm a man. . ., I'm a real Ngoni*'

[*FG8*]

However, the most common reason for the growing substance use problem among youth voiced in all group discussions was that alcohol and drugs were too cheap and very easy to access. Participants felt vendors intentionally turn a blind eye to checking whether customers are of legal age to purchase alcohol or permitted drugs because they prioritise their own goal of profiteering over youth's health and wellbeing. They also felt the government does not do enough to legislate or enforce alcohol and tobacco sales because they too are corrupt and prioritise the income gained from liquor sales.

'*Some people don't want to lose business you know. So, they just let you buy it anyway. Just take. Some youngsters will say 'I've been sent' you know. I've been sent to buy for my father, but you can't really check that. So, I think there needs to be harsher regulation and there should be legal consequences for selling like alcohol and cigarettes to minors.*'

[*FG6*]

**Increasing cases of suicide.**    There were multiple accounts from participants of finding someone or hearing of someone they knew who had committed suicide and reading reports of increasing incidents of suicide. Participants strongly felt that the rise in suicide cases in Malawi is another consequence of the socioeconomic challenges that young people are facing because they feel overwhelmed and experience mental distress. However, because most people in Malawi feel unable to open up and talk about their mental health issues, they remain untreated. As a result, some people resort to taking their life.

'*Here in Malawi, I've never heard someone going for, to meet a psychologist with a problem. That's why we are having a lot of suicide cases in Malawi because last year eeeeh*! *It was too much. I think we need to take it serious.*'

[*FG9*]

**COVID-19 has a negative and positive side.**    Many schools have not been able to provide online learning for students during school closures meaning a lot of young people are at home with nothing to do. For many students, the pandemic has meant a delay in their completion of studies, bringing additional financial burden on families. As one participant stated when talking about lack of things to do during COVID-19, '*An empty mind is a devil's den*' [*FG11*]. Having nothing to do and uncertainties about one's future has consequently increased mental health problems, substance use and risky sexual behaviours among the youth.

'*COVID-19 has planted fear amongst us young people. So that is leading us into depression which is also another issue of mental health since we are just staying idle doing nothing*'

[*FG10*]

'*Just to concur with what has said, you know with this closure of schools, I think we are registering a lot of alcohol and drug abuse, at the same time there is a lot of multiple relationships among the youth*'

[*FG10*]

Then again, a few participants felt that a positive side to the pandemic is that it has allowed people to become more aware of mental health. There was agreement in the discussions that people are being more open about mental health issues, with a lot more conversation around stress, depression, anxiety, and suicide especially among youths on social media. This has been a bittersweet realisation for participants that it took an entire global pandemic for people in Malawi to validate conditions like depression and anxiety as serious issues.

'*a slight positive though to the COVID-19 pandemic and mental health scope. I think over time as the pandemic has gone on we have developed this understanding that we are not alone and that we have this shared traumatic experience, which I think especially for young people is helping them talk about what they are going through more.*'

[*FG1*]

### Theme 3: Relationship issues

**Strained family and community dynamics.**   In addition to the socioeconomic and mental health challenges young people are experiencing as described by participants, inadequate or complete lack of support from parents is another major perceived cause of mental health problems among youth. Participants stated that youths cannot speak freely to their parents about many life struggles like academic pressures, problems with romantic relationships and mental health problems.

'*They didn't have like a good relationship with their parents but then they kept on sucking it in. But then it reached a step at which they couldn't keep it in at all.*'

[*FG3*]

A lot of participants stated that many young people struggle to navigate friendships, romantic relationships, coping with breakups and acquiring sexual health knowledge. However, speaking to parents about sex in Malawi is culturally inappropriate, leading many young people to seek advice from friends, resort to substance abuse, and in extreme cases, commit suicide.

'*A movie, it displays the culture of some people. So, you see that you find the child talks free to their father. If you try to talk about relationships to my father, he will tell you, you are playing my friend which is money, which is my money. I'm paying for your fees.*'

[*FG11*]

Participants also felt that parents are not open with their children when they are going through financial or mental health struggles. This societal behaviour of bottling up problems or dealing with problems alone is also prevalent on a wider family and community level, propagating a toxic culture of suffering in silence and neglecting one's mental health.

'*Within the families and actually the entire society, the relationship is not good and then peo-ple, young people do not have anywhere to go where they can find relief*

[*FG8*]

Participants also voiced frustrations with the government. One focus group felt government youth officers are detached from youth needs, especially existing problems of unemployment, drug and alcohol abuse, mental health, and sexual reproductive health. They voiced frustra-tions of visiting the district youth office multiple times a day and never finding a youth officer to speak to about their concerns. This leaves a lot of pressure on youth advocacy groups that due to limited funding struggle to provide sustained youth initiatives and activities. They described the Government's failure to involve the national youth council during the develop-ment of COVID-19 measures as an example of how little they prioritise young people. Overall, they questioned the use of government led youth organisations.

'*I don't see the activities or the importance of having district youth councils because they are doing nothing, like literally nothing. There is nothing important they are doing. So I think it's high time that now we have to look at what, what the youth councils can be doing so that it can benefit the youths here in Malawi.*'

[*FG9*]

**Societal pressures.** On a broader societal level, participants voiced frustrations regarding societal pressures and expectations that cause young people mental distress. These pressures range from how you dress or present yourself, the type of job you have, getting married before you were too old and having financial independence. In the quote below, a female participant talked about societal pressures that were specific to gender roles

'*Most of the mental issues we have now. . . as Malawians, most of them stem from the culture that we have over here. Because for example for me as a woman, the cultural practices are so sexist. They don't favour me. And then when you are growing up as a girl, I think there was a point when I was younger where wearing a trouser was such a really bad thing*'

[*FG1*]

Participants also described peer pressure among youths themselves which at times causes people to take drugs, abuse alcohol or be involved in risky sexual practices just to fit in with cer-tain friendship circles and trendy practices. This is a particular issue in university settings where students from different social backgrounds try to fit in with more elite friendship circles.

'*FG11-P1: Girls at college tend to emulate the behaviour of others coz maybe they just want to be recognized that they are at school. So, we have seen girls at school, where basically it's called a G-party, maybe can be my witness [FG11-P2: Yeah, g-party, gang rape] whereby you see a girl, a girl sleeping with more than one person at one time.*'

[*FG11*]

The strained relationship between young people and their parents or guardians, plus lacking mentorship within community pushes them to be over dependent or trusting of their friends for support and guidance on life challenges, regardless of reliability of these friendships.

'*If I have friends that usually smoke, I would end up start smoking because they are friends. They could actually convince me to say, lets smoke and blow ourselves high'. I'd listen to them because they are friends and I'd put my trust in them.*'

[*FG9*]

## Theme 4: Lacking mental health support

Participants in all nine focus groups expressed lacking mental health support for young people and more broadly for Malawians. As detailed later, they described (i) a lack of mental health institutions people could go to for treatment, (ii) a lack of qualified mental health workers and (iii) a lack of knowledge about mental health within the country population that affected people recognising when they were ill, plus when and where to go for help.

'*Mama this is Malawi. We are not in the UK. We are not in South Africa. It's Malawi. Practically speaking there's nowhere. You cannot go anywhere. I don't know where to go. If I was to say that I'm depressed. I'm having problems from my kids. My kids are giving me hell, I want to get out of this place. I don't know where to go. My room. Yes, run to my room*'

[*FG6*]

**Service provision and workforce.** Participants felt the mental health system was severely underfunded by government and there were very few mental health organisations which is a key barrier to young people accessing much needed support.

'*That is only one for the whole northern region offering professional mental health counselling and for the other central and southern region, there are three.*'

[*FG9*]

Existing structures were hospital structures more geared for severe psychiatric cases and not youth friendly which was a barrier to access. Another barrier participants noted was that there were very few qualified mental health practitioners who could provide young people with mental health support when needed. Because there are very few existing practitioners, therapy in Malawi is very classist, only available to those who can afford to pay expensive fees for private sessions.

'*Another problem is that one of lack of psychologists everywhere because we have schools, we have primary schools because I would prefer to say maybe we have an "in-house" psychologist in each and every school so that they can be monitoring the mental health of the students and also responding to the needs of their mental health.*'

[*FG8*]

**Sources of mental health information and support.** Most participants said they got mental health information from the internet on social media and WhatsApp groups. However, they generally felt it was difficult in Malawi to access mental health information. Young people from rural communities or without access to internet would be particularly disadvantaged.

'*Majority of the young people in Malawi, they don't have really a place where they will go to get information.*'

[*FG8*]

Some participants mentioned a few organisations that provide psychosocial services as an information point.

'*The one-stop centre at Queens. They work with mental health issues in children. They say teenagers come from 18 below and those who were rape victims or maybe even drug abuse, maybe a problem child.*'

[*FG2*]

Due to lacking access and provision of mental health support, most people seek help from alternative sources like church where they can find emotional and practical support from within a community they are already connected to.

'*The pastor who I was talking about. He is a good friend whenever, because he knows of, he will tell you about threats and he will also tell you about what is really happening in this world, about mental health.*'

[*FG11*]

Friends more than family were also a common alternative source of support for young people.

'*I have a few friends that I can go to, and I can be open with. Yeah, mostly it's friends, family aaaah* [*doubtfully*], *I don't think I can open up to them.*'

[*FG11*]

Participants stated that low mental health literacy in Malawi among adults and youths as another key barrier to young people having access to mental health support. Several of them admitted learning about mental health during late teens or whilst at university during discussions which they felt was too late. This lack of knowledge and awareness within the country means stigma towards mental health is rife which further hampers help seeking.

'*People don't want to go there because of stigma issues. I think people now have this acceptance thing that when someone is mentally disturbed, they are mentally disturbed. We will be living with them in our homes until they can get better. And that's just it.*'

[*FG2*]

## Suggested mental health solutions

Participants strongly felt that raising awareness of mental health in Malawi should be prioritised. A common suggestion was targeting young people within educational settings in the form of ad-hoc workshops, talks, drama, and as part of core curriculum. As illustrated in the quote below, participants felt sharing lived experiences could be particularly effective.

'*It's not easy going to colleges where you see mental health literacy it's not good. So, you need to have people who went through mental health, and they are now back to normal. It's a testimony. Usually, it carries weight to people's life.*'

[*FG11*]

Integrated psychologist or therapist on site in schools to ensure there is consistent support was another solution. Considering the countrywide shortage of psychologists, participants felt this could be a job that social work or teaching graduates could do if they were given the appropriate training. Participants also felt that more mental health promotion needed to be done in the wider community, beyond schools and young people. Suggested activities within these settings included door to door needs assessment, focus groups divided by family, gender and ages, as well as dramas. Recognising that mental health funding in Malawi was skeletal, participants suggested the use of existing structures within localities like youth centres, churches and district council offices. A key approach was to gain buy-in from community leaders like chiefs on any proposed initiative so that they could share this with their communities and encourage engagement. Moreover, chiefs would be vital in informing researchers of the needs of their community, and this would be useful to tailor content.

'*When you talk about the communities where we can have these focus groups, we have the chiefs in our communities. So, I think those ones can give us the best places where we can be connecting.*'

[*FG6*]

There was also great need to have more local, youth friendly services to narrow-down barriers to access of mental health services like transport costs.

'*due to issues of poverty, transport. They should travel maybe from Zalewa to get to Blantyre city to access that (counselling). Come back and forth so that they should have so many sessions but maybe those people don't have that kind of money. How about for each and every community to at least have an office so that people can at least just use their legs to walk there and access their services.*'

[*FG2*]

Participants also suggested coming up with a mental health advocacy strategy with local and central government, along with policy makers to increase literacy within communities on what mental health is, its effects and impacts. They also felt using performing arts on (social) media outlets/platforms could be a particularly effective way to disseminate mental health information to young people.

'*We also need strong advocacy strategies that we can work on coz in mental health here in Malawi, we are very far. Many people in Malawi, they don't understand the concept of mental health*'

[*FG10*]

## Discussion

This study adds to the limited body of qualitative research in Malawi, by uniquely exploring mental health challenges that young people face, from their perspectives and lived experiences.

The novelty of this work was the proactive use of CE to obtain views of relevant grassroots youth groups to inform the development of mental health interventions for our specific target population. Another novel aspect was the use of social media to recruit participants and conducting the focus groups online. We expect this approach will yield relevant and effective solutions for young people in Malawi and their wider community.

Depression and anxiety, substance use and suicide were key health concerns voiced by all study participants alongside recognition of the need for interventions to address the identified challenges. Socioeconomic challenges and key drivers of poverty like youth unemployment and financial problems were the main causes of identified key health concerns. Moreover, a neglected mental health system and lacking support from parents and wider community left these mental health and related concerns unaddressed. Participants felt using existing community structures to increase access to mental health support was a good way to strengthen Malawi's mental health system and crucially address the mental health needs of youth and broader communities. Raising awareness, capacity building, integration of mental health screening and support within schools and local communities were recommended tools to address youth mental health issues.

### Experiences of young people with mental health problems

There is growing evidence highlighting the magnitude and rising prevalence of mental disorders, mental health symptoms and substance use among youth in Africa [2, 8, 9, 28]. More recently suicide is becoming another growing topic of concern [29, 30]. COVID-19 has amplified these existing problems for young people on a national, regional, and global scale [31, 32]. Our study corroborates these findings. Like other studies, cannabis and alcohol are common substances of abuse among youth in Malawi [8], although there was also reported evidence of novel illicit drugs and concoctions in more urban areas. Participants pointed out that people in Malawi sometimes found it hard to find relevant phrases to express or describe their experiences of distress in ways applicable to the western concept of mental health. This issue of differing cultural concepts and language used within mental health has been similarly reported in other studies conducted in Malawi and elsewhere [33–36]. This shows the importance of identifying terminology for terms like mental health, depression, and anxiety that local communities in Malawi can connect with [36]. It is also very important to incorporate appropriate language in mental health services and research alike to facilitate more culturally relevant and effective treatments and interventions and minimise health disparities [7, 34, 35, 37].

Relationship problems with primary caregivers are also known to negatively impact on a young person's mental health / suicidal ideation. Similarly, with regards to substance use, studies in Africa indicate the parenting style/ issues with parents/ broken families are a risk factor for drug and alcohol abuse in young people. Young people felt that people with conditions like mental health continue to face stigma in society which made them fearful to disclose psychological distress to others [15]. As such, it appears mental health stigma towards individuals and their families remains rampant in Malawi [15, 16, 38] and the broader region [39] which negatively impacts help seeking and treatment.

### Service needs

Our study shows that awareness raising and education on mental health is vitally important for young people [1]. Targeting their parents and the wider society in Malawi is also key so that they are better able to support young people who are at risk of mental illness [15, 38, 39]. The young people we spoke to outlined inadequacies in information and support available concerning youth mental health. This was compounded by a weak mental health system,

lacking availability of health workers and negative attitudes towards mental health within the society [1, 7, 10]. While shortage of health workers, poor access to facilities and negative attitudes have been found to be major barriers to accessing mental health treatment in LMICs [7, 28, 38], and lack of mental health knowledge is an additional factor that hampers provision of support to individuals in need [1, 38]. Certain cultural practices in Malawi where families tend to carry the caring burden also delay help seeking [40]. Increasing mental health literacy on both individual and societal levels would improve recognition, help seeking and management of mental illness [1].

We have used lessons learnt from this study to modify content from a Canadian school-based mental health literacy resource adapted for application in sub-Saharan Africa [1, 17] into an online curriculum for university students in Malawi. Examples include adding digital stories from Malawians with lived experiences of mental health and signposting to existing local service providers that provide psychosocial support to young people in Malawi. We hope these changes enhance cultural relevance of the curriculum content and the online format will minimise disruption of delivering the curriculum during future COVID-19 outbreaks. We are piloting delivery in a Malawi university, assessing potential impact of this curriculum on students' literacy. We have also conducted a national survey with young people aged 16 to 30 years in rural and urban settings of Malawi to gauge baseline mental health literacy [41] as this can help inform how to target and pitch content during future mental health promotion activities. Pilot study outcomes regarding the online curriculum's impact and the mental health literacy survey will be published separately. As a next step, we plan to implement this online mental health literacy curriculum on a larger scale within a feasibility trial. The trial will run alongside community engagement activities e.g., radio interviews and debates on radio and social media with young people and their peers to promote mental health discourse beyond universities.

## Strengths and limitations

Study strengths include recruiting from a broad spectrum of young people based in educational settings and/or existing youth advocacy structures. Using Twitter and Facebook for participant recruitment aptly suited our target group who spend a lot of their time on social media platforms. Conducting online focus groups removed geographical restrictions, allowing us to reach participants nationwide. Both these approaches to study recruitment are novel for the Malawi setting and worked well with our target population, providing scope for future use in research with young people. This study is one of few in Malawi reporting findings from CE experiences [19] and the first using this approach as a tool to inform youth mental health literacy intervention development. Insights from these focus groups have been used to inform content of our e-curriculum, for example signposting youths to local services and the creation of digital stories on living with mental disorders. We have also learnt about contextual factors that could impact implementation of interventions targeting youth in Malawi. Examples include the high cost of internet and limited access to devices like laptops. This highlights the importance of multimodal approaches to intervention delivery, i.e., both digital and face to face, and need of linking with grassroots community leaders for extensive outreach of mental health promotion.

Limitations include conducting online focus groups which excludes those experiencing digital poverty. For example, six individuals who were very eager to participate were unable to join. We subsequently offered them to do a telephone interview instead. More broadly, most Malawian youth live in rural areas and may not have access to social media or internet so they would not have seen the study advertisement. As perceptions of such individuals were not included, our findings may not be typical for all young people. However, we did have several participants who grew up, lived, or worked in rural communities.

## Conclusion

Our study revealed the complex health, psychosocial, social and economic issues affecting young people's mental health in Malawi. Alongside these issues, challenges in the country's health system that hinder mental health support services specifically for youth were highlighted. This work provides support for plans to adapt existing or develop new interventions that focus on the articulated needs of youth, their families, and communities. It specifically calls for a need to raise awareness of mental health not just among youth but their caregivers, organisations like educational institutions and churches and wider community. Benefits of such interventions could be early recognition, improved help seeking and better self-management of mental health in the short term. If our e-curriculum is rolled out nationally, longer term impacts may be increased mental health literacy on a population level alongside alleviating pressure on an already weak health system and overworked medical staff. Finally, mental health services need to be strengthened and appropriately tailored for different demographics and use existing community structures to increase accessibility for young people.

## Supporting information

**S1 Fig. Focus group discussion guide.**
(DOCX)

## Acknowledgments

We sincerely thank all the young people and community youth groups for spending their time to provide their invaluable insights during the focus group discussions and subsequently as the intervention was being developed. We also thank Dr Ukwuori Kalu for reviewing our manuscript from a specialist children and young persons' mental health service perspective.

## Author Contributions

**Conceptualization:** Sandra Jumbe, Gase Motshewa, Subba Rao Pulapa.

**Data curation:** Sandra Jumbe, Joel Nyali, Gase Motshewa.

**Formal analysis:** Sandra Jumbe, Joel Nyali.

**Funding acquisition:** Sandra Jumbe.

**Investigation:** Sandra Jumbe, Joel Nyali, Maryrose Simbeye, Nelson Zakeyu.

**Methodology:** Sandra Jumbe, Gase Motshewa, Subba Rao Pulapa.

**Project administration:** Sandra Jumbe, Gase Motshewa.

**Resources:** Gase Motshewa.

**Software:** Sandra Jumbe, Gase Motshewa.

**Supervision:** Sandra Jumbe, Subba Rao Pulapa.

**Validation:** Sandra Jumbe, Subba Rao Pulapa.

**Writing – original draft:** Sandra Jumbe.

**Writing – review & editing:** Sandra Jumbe, Joel Nyali, Maryrose Simbeye, Nelson Zakeyu, Gase Motshewa, Subba Rao Pulapa.

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
