## [Decision Letter · Decision Letter 0]

28 Jan 2022

PONE-D-21-29945‘We do not talk about it’: Engaging youth in Malawi to inform adaptation of a mental health literacy interventionPLOS ONE

Dear Dr. Jumbe,

Thank you for submitting your manuscript to PLOS ONE. After careful consideration, we feel that it has merit but does not fully meet PLOS ONE’s publication criteria as it currently stands. Therefore, we invite you to submit a revised version of the manuscript that addresses the points raised during the review process.

We look forward to receiving your revised manuscript.

Kind regards,

Sonia Brito-Costa, Ph.D.

Academic Editor

PLOS ONE

Journal Requirements:

Reviewers' comments:

Reviewer's Responses to Questions

5. Review Comments to the Author

Reviewer #1: Thank you for giving me the opportunity to review this manuscript. It is very interesting, and quite up to the date. I have some comments and doubts for the authors.

First of all, the population interviewed are universitary people, why?

Refereing to the sample, the authors said that 76 people agreed to participate; 44 of them enroled into the interviews, 25 did not give an answer and 6 had connectivity problems... That sum reaches to 75 people. There ir one missing person.

How did the authors coded the topics and the subtopics? Which was the followed procedure?

It would have been really interesting to also know the proposals of the youth in regards to mental health improvement, in order to stablish future research lines.

What are the main objectives of these results obtained? Which is the next step related to the improvement of youth Malawis mental health?

---

## [Author Response · Author response to Decision Letter 0]

5 Feb 2022

To: Dr Emily Chenette 

Editor-in-chief, PLOS One 

3rd February 2022 

Dear Dr Chenette and associated PLOS One editors 

RE: Manuscript submission reference PONE-D-21-29945 

Many thanks for the opportunity to revise our manuscript. You will find attached clean and track change revisions of our manuscript, “‘We do not talk about it’: Engaging youth in Malawi to inform adaptation of a mental health literacy intervention”. We indicate below how we have addressed comments from reviewer #1. 

1. Thank you for giving me the opportunity to review this manuscript. It is very interesting, and quite up to the date. I have some comments and doubts for the authors. 

Response: Thank you for your positive feedback, acknowledging interest of this paper, especially how it addresses current issues regarding youth mental health in Malawi. We have addressed your comments below and revised our manuscript accordingly 

2. First of all, the population interviewed are universitary people, why? 

Response: Thank you for your question. The population interviewed was not just university people. As outlined in the abstract and the ‘study setting’ paragraph under the ‘Materials and Methods’ section, the sample included young people from universities and other community settings e.g. youth organisations in Malawi (see page 6 line 3-9). 

3. Refereing to the sample, the authors said that 76 people agreed to participate; 44 of them enroled into the interviews, 25 did not give an answer and 6 had connectivity problems... That sum reaches to 75 people. There ir one missing person. 

Response: Thank you for noting this typing error. It was 75 people who agreed to participate initially. This has been amended on the manuscript under the ‘participant recruitment’ section (page 6 line 25) 

4. How did the authors coded the topics and the subtopics? Which was the followed procedure? 

Response: We used qualitative content analysis as an approach to analysis data from the focus group transcripts. The procedure followed to code the themes and subthemes is detailed in the ‘Data collection, analysis, and management’ section. We have revised this section with more details and an additional Table 3 (see page 9) to clarify this process (see page 8 line 13 to page 9 line 21). We have also included a paragraph on rigor of the data analysis process (page 10 line 5 to page 11 line 13) 

5. It would have been really interesting to also know the proposals of the youth in regard to mental health improvement, in order to stablish future research lines. 

Response: Thank you. This has already been outlined in the results under the section ‘suggested mental health solutions’ (see page 27 line 5 to page 28 line 20), where the youth proposed several ideas regarding mental health improvement. Examples in this section include more awareness raising of mental health, having a mental health practitioner within schools for consistent access to psychological support for students, creating local youth-friendly mental health services and top-down mental health advocacy strategy from government to facilitate mental health literacy in communities countrywide 

6. What are the main objectives of these results obtained? Which is the next step related to the improvement of youth Malawi's mental health? 

Response: Thank you for your comment. We have added details on work we have done and related next steps to the discussion section (see page 31 line 7-24) 

We have also addressed additional comments from the editor as outlined below: 

1. We have revised headings, tables and figures labels in the manuscript in line with the PLOS ONE style templates found at 

2. In line with PLOS ONE data availability guidance, we have uploaded the study’s focus group discussion guide as the minimal anonymized data set to facilitate replication of our study findings as a supporting Information file (S1 Fig on page 33 line 18). Please update our Data Availability statement on our behalf to reflect the information we have provided. 

3. We have uploaded our 2 figure files to the Preflight Analysis and Conversion Engine (PACE) digital diagnostic tool and inspected them for image clarity and content. 

Addressing the above comments has improved our manuscript and we are confident that the paper now meets PLOS ONE’s publication criteria. We look forward to your favourable opinion on our revised manuscript. 

Yours sincerely 

Dr Sandra Jumbe

---

## [Decision Letter · Decision Letter 1]

22 Feb 2022

PONE-D-21-29945R1‘We do not talk about it’: Engaging youth in Malawi to inform adaptation of a mental health literacy interventionPLOS ONE

Dear Dr. Jumbe,

Thank you for submitting your manuscript to PLOS ONE. After careful consideration, we feel that it has merit but does not fully meet PLOS ONE’s publication criteria as it currently stands. Therefore, we invite you to submit a revised version of the manuscript that addresses the points raised during the review process.

We look forward to receiving your revised manuscript.

Kind regards,

Sonia Brito-Costa, Ph.D.

Academic Editor

PLOS ONE

Journal Requirements:

Reviewers' comments:

Reviewer's Responses to Questions

**Comments to the Author**

Reviewer #1: Thank you for consider my previous comments.

The only issue still concerns me is related to qualitative statistical analysis. Now is more complete and clear, but, did the authors use any statistical program to conduct these analyses, such as Atlas.ti or similar? If not, why?

Thank you for your effort and your job.

---

## [Author Response · Author response to Decision Letter 1]

2 Mar 2022

To: Dr Emily Chenette 

Editor-in-chief, PLOS One 

2nd March 2022 

Dear Dr Chenette and associated PLOS One editors 

RE: Manuscript submission reference PONE-D-21-29945R1 

Many thanks for the opportunity to revise our manuscript. You will find attached clean and track change revisions of our manuscript, “‘We do not talk about it’: Engaging youth in Malawi to inform adaptation of a mental health literacy intervention.” We indicate below how we have addressed the comment from reviewer #1. 

Reviewer 1: Thank you for consider my previous comments. The only issue still concerns me is related to qualitative statistical analysis. Now is more complete and clear, but, did the authors use any statistical program to conduct these analyses, such as Atlas.ti or similar? If not, why? Thank you for your effort and your job. 

Thank you for acknowledging our efforts in addressing the original comments from reviewers to our manuscript. We are glad that you feel the manuscript is now complete and clear. Regarding your question, we already stated in the manuscript on page 8 lines 12-15 that we used NVivo to conduct our qualitative analysis. We have added a description of what NVivo is on page 8 lines 15-17 to make this aspect clearer. 

Addressing the above comment has further clarified the data analysis section of our manuscript and we are confident that the paper now meets PLOS ONE’s publication criteria. We look forward to your favourable opinion on our revised manuscript. 

Yours sincerely 

Dr Sandra Jumbe

---

## [Editor Report · Decision Letter 2]

4 Mar 2022

‘We do not talk about it’: Engaging youth in Malawi to inform adaptation of a mental health literacy intervention

PONE-D-21-29945R2

Dear Dr. Jumbe,

We’re pleased to inform you that your manuscript has been judged scientifically suitable for publication and will be formally accepted for publication once it meets all outstanding technical requirements.

Kind regards,

Sonia Brito-Costa, Ph.D.

Academic Editor

PLOS ONE
---

## [Editor Report · Acceptance letter]

21 Mar 2022

PONE-D-21-29945R2 

*‘We do not talk about it’: *Engaging youth in Malawi to inform adaptation of a mental health literacy intervention 

Dear Dr. Jumbe:

I'm pleased to inform you that your manuscript has been deemed suitable for publication in PLOS ONE. Congratulations! Your manuscript is now with our production department. 

Kind regards, 

on behalf of

Dr. Sonia Brito-Costa 

Academic Editor

PLOS ONE